# Cough Characteristics and Their Association Patterns According to Cough Etiology: A Network Analysis

**DOI:** 10.3390/jcm12165383

**Published:** 2023-08-18

**Authors:** Jieun Kang, Ji-Yong Moon, Deog Kyeom Kim, Jin Woo Kim, Seung Hun Jang, Hyeon-Kyoung Koo

**Affiliations:** 1Division of Pulmonary and Critical Care Medicine, Department of Internal Medicine, Ilsan Paik Hospital, Inje University College of Medicine, Goyang 10380, Republic of Korea; realodette@gmail.com; 2Department of Internal Medicine, Hanyang University College of Medicine, Guri 11923, Republic of Korea; respiry@gmail.com; 3Division of Pulmonary and Critical Care Medicine, Department of Internal Medicine, Seoul Metropolitan Government-Seoul National University Boramae Medical Center, Seoul National University College of Medicine, Seoul 07061, Republic of Korea; kimdkmd@snu.ac.kr; 4Division of Pulmonary and Critical Care Medicine, Department of Internal Medicine, Uijeongbu St. Mary’s Hospital, College of Medicine, The Catholic University of Korea, Uijeongbu 11765, Republic of Korea; medkjw@catholic.ac.kr; 5Division of Pulmonary, Allergy, and Critical Care Medicine, Department of Medicine, Hallym University Sacred Heart Hospital, Hallym University College of Medicine, Anyang 14068, Republic of Korea; chestor@hallym.or.kr

**Keywords:** chronic cough, COugh Assessment Test, etiology, network analysis

## Abstract

Although cough is a common respiratory symptom, determining its cause is challenging. We aimed to explore how cough severity and characteristics vary with different etiologies, while investigating their interrelations with demographic features. Adult patients (*n* = 220) with chronic cough and completed diagnostic work-up and the COugh Assessment Test were enrolled. A correlation network analysis was used to examine the associations between the demographic features and cough severity/characteristics across various etiologies such as upper airway cough syndrome, asthma, eosinophilic bronchitis, gastroesophageal reflux disease (GERD), and idiopathic cough. Demographic features like age and sex showed complex associations with cough characteristics and severity. Cough severity decreased with age, especially in cases of eosinophilic bronchitis and GERD. Women with eosinophilic bronchitis reported more severe cough, while men with idiopathic cough reported more severe cough. Asthma was significantly linked to more sleep disturbance and fatigue, independent of age and sex, whereas GERD showed less sleep disturbance and fatigue. Network analysis revealed overall close associations between cough characteristics, though hypersensitivity in asthma and sleep disturbance in GERD were not linked with other cough traits. In conclusion, the demographic features and cough characteristics were interrelated, exhibiting distinct patterns based on the etiology.

## 1. Introduction

Cough is a common respiratory symptom that causes patients to seek medical care [1]. Although cough is an important defense mechanism that protects the airway from foreign bodies and clears excessive secretions, it can cause discomfort and impair the patients’ quality of life, especially when it becomes chronic [2,3]. Identifying the cause of chronic cough can be difficult because it may be multifactorial, and causative symptoms such as postnasal drip or gastroesophageal reflux are often clinically silent [4].

Upper airway cough syndrome (UACS), asthma (including cough variant asthma), eosinophilic bronchitis, and gastroesophageal reflux disease (GERD) have been emphasized as the major causes of chronic cough in non-smokers, where chest radiographs are normal [5,6]. It is largely unknown whether the severity or characteristics of cough differ between diseases. The detailed evaluation of cough characteristics may help elucidate the mechanism by which cough develops in each disease, and ultimately identify the cause of chronic cough from the symptom itself.

Cough questionnaires are tools that not only measure cough severity, but also evaluate the patients’ quality of life in terms of various aspects [7]. One such example is the Leicester Cough Questionnaire [8]. The COugh Assessment Test (COAT) is a simplified version of the Leicester Cough Questionnaire, which has been validated as a useful tool for assessing and monitoring cough, showing good correlations with the Korean version of the Leicester Cough Questionnaire (K-LCQ) and the cough numeric rating scale (NRS) [9]. The COAT consists of items that assess five cough characteristics. The total and individual item scores represent the overall cough severity and the degree of each characteristic, respectively.

We aimed to evaluate whether the severity and characteristics of cough differed according to etiology and investigate how cough severity, characteristics, and demographic features such as age and sex are interrelated.

## 2. Materials and Methods

### 2.1. Study Patients and Data Collection

Data of adult patients aged ≥ 18 years with chronic cough were retrospectively collected from 16 respiratory centers in the Republic of Korea from 1 March 2016 to 28 February 2018. Chronic cough was defined as a cough persisting for longer than 8 weeks [10]. The possible causes of chronic cough were assessed by pulmonary specialists at each hospital following the diagnostic algorithm suggested by the Korean Cough Guidelines [10]. Patients without suspected abnormalities on chest radiographs and those who completed the COAT questionnaire were included. Patients were excluded from the analysis if they did not finish the diagnostic work-up to determine the cause of chronic cough, or had multiple causes for chronic cough. Those with a known chronic respiratory disease such as overt asthma, chronic obstructive pulmonary disease, bronchiectasis, tuberculosis-destroyed lung, or lung cancer were also excluded.

### 2.2. Cough Evaluation Methods

The K-LCQ is a cough-specific questionnaire for quality of life validated in Korean patients, consisting of 19 items of physical, psychological, and social domains. Each item is scored by a 7-point Likert scale, ranging from 1 to 7, and total scores are produced by the summation of the calculated mean scores for each domain [8]. A higher K-LCQ score indicates a better quality of life. The COAT is a validated simplified version of the K-LCQ, which contains five items: frequency of cough, limitation of daily activities, sleep disturbance, fatigue, and hypersensitivity to irritants. Each item was considered a distinct cough characteristic impacting the health-related quality of life. All items were scored on a single scale ranging from 0 to 4, constituting a total score ranging from 0 to 20, with higher scores indicating more severe cough symptoms [9]. The cough NRS ranges from 0 (no cough at all) to 10 (maximal cough).

### 2.3. Ethical Statement

The authors are accountable for all aspects of the work in ensuring that questions related to the accuracy or integrity of any part of the work are appropriately investigated and resolved. This study was conducted in accordance with the Declaration of Helsinki (as revised in 2013). The study was approved by the Institutional Review Board of Ilsan Paik Hospital (No. 2017-12-025) and the need for informed consent was waived due to the retrospective nature of the study.

### 2.4. Statistical Analysis

Data are presented as the means and standard deviations for continuous variables and as numbers (%) for categorical variables. For continuous variables, the Student’s *t*-test was used to compare groups. Categorical variables were compared using the chi-square or Fisher’s exact test. All statistical analyses were performed using R software (version 3.6.0, R Foundation for Statistical Computing, Vienna, Austria). A multivariate analysis for each etiology, using cough severity and characteristics, was performed using logistic regression adjusted for age and sex. A correlation matrix was constructed using the cor function for Pearson’s correlation. The Pearson’s correlation coefficient, a value ranging from −1 to 1, quantifies the normalized covariance between two variables. The degree of correlation was defined by Pearson’s co-efficient, as strong (>0.6), moderate (0.3–0.6), or weak (<0.3) [11]. A correlation network was drawn based on the correlation matrix, and each variable was represented as a specific node [12]. The degree of correlation was defined by Pearson’s coefficient: >0.6 as strong, 0.3–0.6 as moderate, and <0.3 as weak. A correlation network was drawn based on the correlation matrix, and each variable was represented as a specific node. The size of the node indicated the prevalence: patients with an age > 65 years, women, and the relative COAT scores that were calculated by the mean of each COAT item score divided by the total COAT score. Links or edges between nodes indicated statistically significant associations (*p* < 0.05). The thickness of an edge represented the strength of the correlation (Pearson’s R coefficient), and blue and pink indicated the positive and negative correlations, respectively. The igraph package was used to visualize the correlation networks.

## 3. Results

### 3.1. Baseline Characteristics of Study Patients

A total of 322 adult patients with chronic cough were enrolled from 16 respiratory centers in the Republic of Korea. Among them, 255 patients who completed the diagnostic work-up to confirm the etiology and the COAT questionnaire were included. Thirty-two patients with two or more causes of chronic cough were excluded. Among the rest, three patients in whom smoking was the cause of chronic cough were excluded due to the small sample size, resulting in 220 patients being included in the final analysis.

The baseline characteristics of the study patients and their chronic cough etiologies are presented in Table 1. The mean age was 47.9 ± 14.1 years, and women accounted for 64.6% of the patients. The most common cause of chronic cough was UACS (42.6%), followed by asthma (22.9%) and eosinophilic bronchitis (16.6%). GERD and idiopathic cough accounted for 9.4% and 7.2% of the cases, respectively.

### 3.2. The Cough Severity and Characteristics According to Etiology

The overall severity of cough, represented by the total COAT score, differed depending on the etiology (Table 1). The total COAT score was the highest for asthma, followed by UACS and idiopathic cough. The cough severity in GERD was the lowest. Using the K-LCQ and NRS, cough severity was also the highest in asthma and the lowest in GERD.

The cough characteristics differed for each disease. Sleep disturbances were most severe in patients with asthma, followed by UACS. Patients with asthma also reported the highest fatigue score, whereas those with idiopathic cough reported the most severe symptom of hypersensitivity to irritants. Patients with GERD showed the lowest degree of symptoms for all five items in the COAT questionnaire.

### 3.3. Impact of Age on Cough Severity and Characteristics

Figure 1 shows the effect of age on the total COAT score. Increasing age was associated with a lower cough severity (regression coefficient, −0.065; *p* < 0.001). This correlation was more prominent in men (regression coefficient, −0.103; *p* = 0.002) than in women (regression coefficient, −0.053; *p* = 0.038).

Table 2 shows the correlations between age, sex, and cough severity and characteristics by the COAT score. Age was not only negatively correlated with severity but also with cough characteristics such as frequency of cough, limitation on daily activities, fatigue, and hypersensitivity to irritants. The COAT items were associated with each other in moderate to high degrees, except for a weak association between sleep disturbance and hypersensitivity.

The age distribution of the study patients with each cough etiology is shown in Appendix A. The association of age on the severity of cough differed according to the cough etiology (Figure 2). In eosinophilic bronchitis and GERD, increasing age was significantly associated with a lower total COAT score, whereas this association was not found in UACS, asthma, and idiopathic cough. When analyzed with K-LCQ scores instead of the COAT, the results were similar; increasing age was significantly correlated with better quality of life in eosinophilic bronchitis and GERD (Appendix A). Using the NRS, increasing age was also shown to be associated with less severe cough in GERD (Appendix A).

### 3.4. Impact of Sex on Cough Severity and Characteristics

Sex also had different effects on cough severity according to the etiology of cough, although the cough severity did not differ between men and women (Table 3). Men had significantly higher total COAT scores than women with idiopathic cough, whereas women had significantly higher total COAT scores than men with eosinophilic bronchitis. Depending on the cough etiology, there were differences in the cough characteristics between men and women. In asthma and eosinophilic bronchitis, women showed significantly higher fatigue scores than men; women with eosinophilic bronchitis also showed significantly higher hypersensitivity scores. In contrast, cough frequency, limitation of daily activities, and hypersensitivity were higher in men than in women with idiopathic cough. 

### 3.5. Multivariate Analysis of Cough Severity and Characteristics

Table 4 shows the results of the multivariate analysis for each etiology, adjusted for age and sex. Asthma was significantly associated with increased sleep disturbance (odds ratio [OR] = 1.70), fatigue (OR = 1.66), and cough severity (OR = 1.16). In contrast, GERD was significantly associated with less sleep disturbance (OR = 0.51), fatigue (OR = 0.66), and lower cough severity (OR = 0.87). 

### 3.6. Network Analysis

To better understand the complex inter-relationships between age, sex, cough severity, and characteristics using the COAT score for each cough etiology, correlation networks were drawn, as shown in Figure 3. In UACS, the limitations on daily activities decreased as the patient’s age increased. Age was not significantly associated with cough characteristics in patients with asthma or idiopathic cough. In eosinophilic bronchitis, increasing age was associated with less limitation of daily activities, fatigue, and hypersensitivity to irritants. In GERD, older age was associated with a lower cough frequency, less limitation in daily activities, and less hypersensitivity.

Female sex was positively correlated with fatigue in both asthma and eosinophilic bronchitis, and with hypersensitivity in eosinophilic bronchitis; in contrast, negative correlations were found between female sex and cough frequency, daily activity limitation, or hypersensitivity in idiopathic cough. The impact of sex was stronger in idiopathic cough than in other etiologies (Appendix A).

Close association patterns between cough characteristics were observed in patients with UACS, eosinophilic bronchitis, and idiopathic cough. However, the patterns appeared to be different in patients with asthma and GERD. In patients with asthma, hypersensitivity to irritants was only weakly associated with fatigue and was not associated with other characteristics. In GERD, sleep disturbance was only associated with fatigue, but not with other characteristics.

Figure 4 shows the correlation networks for the relationship between age, sex, and the K-LCQ items for each cough etiology. When assessed using the K-LCQ, the effects of age on the various cough characteristics were most prominent in eosinophilic bronchitis and GERD, similar to the results evaluated by the COAT. Female sex was significantly associated with a greater level of tiredness in eosinophilic bronchitis and less anxiety in idiopathic cough.

## 4. Discussion

This study described and compared the severity and characteristics of chronic cough according to etiology by using the COAT questionnaire. Cough was the most severe in asthma and the least severe in GERD. Demographic features such as age and sex had complex associations with the characteristics and severity of cough. The severity of cough decreased with increasing age, especially in cases of eosinophilic bronchitis and GERD. Overall, cough severity did not differ between men and women, but women reported severe cough in eosinophilic bronchitis, whereas men reported more severe cough in idiopathic cough. In the multivariate analysis, asthma was significantly associated with more sleep disturbance and fatigue, independent of age and sex, whereas GERD was associated with less sleep disturbance and fatigue. In general, there were close relationships between all cough characteristics; however, hypersensitivity and sleep disturbance were isolated characteristics in asthma and GERD, respectively.

Previous studies on chronic cough have mainly focused on building a diagnostic algorithm or reporting rare causes of chronic cough [10,13,14,15]. Considering the results of our study, the COAT may help differentiate the cause by evaluating the cough symptom itself. For instance, patients with asthma had the highest cough severity and more sleep disturbance and fatigue than others. This suggests that patients with chronic cough that causes severe impairment in various aspect of their quality of life should be suspected of having asthma. Patients with GERD appear to have a relatively higher quality of life than those with asthma. Meanwhile, symptoms of hypersensitivity to irritants that are more pronounced than other symptoms may suggest idiopathic cough. More importantly, our approach, which assesses cough severity and characteristics according to etiology, may enable us to better understand the pathophysiology of cough development and progression in different diseases. Eosinophilic bronchitis is considered similar to asthma in terms of its eosinophil-driven pathophysiology [16,17]. However, according to our study results, unlike asthma, eosinophilic bronchitis did not have a significant impact on cough scores in the multivariate analysis. The decrease in cough severity with increasing age was similar to that observed in GERD. Eosinophilic bronchitis also showed a different pattern from asthma in terms of the association between cough characteristics in the network analysis. These findings suggest that cough may involve different mechanisms in eosinophilic bronchitis and asthma.

A new insight from our study is that cough severity in UACS should not be underestimated. Cough was the second most severe in UACS. UACS is the most common cause of chronic cough and is often regarded as a benign, non-severe disease [14,18]. However, the quality of life impairment caused by UACS was not negligible, according to the results of our study. More attention to the patients’ quality of life may be required when treating patients with UACS. Meanwhile, we found no clear difference in cough hypersensitivity between men and women, although there have been some controversies regarding cough hypersensitivity in women [19]. Women complained of more fatigue and hypersensitivity in eosinophilic bronchitis, but less hypersensitivity in idiopathic cough.

A notable finding of our study was the association between age and cough severity. Although the prevalence of chronic cough may vary according to age, the subjective perception of severity decreases in many diseases, especially in those presenting with less severe cough. For instance, even though the prevalence of GERD might rise with increasing age [20], it has been suggested that older individuals with GERD experience a dissociation between symptom severity and the extent of underlying esophageal damage [21]. Our analysis also indicated a diminishing trend in cough severity with advancing age within the context of GERD. Age was correlated with various cough characteristics including cough frequency, limitation of daily activities, fatigue, and hypersensitivity, which suggests a possible influence of age on the generation or perception of cough. In general, it is believed that the cough reflex may be decreased in patients with degenerative diseases [22,23] and in the older population [24,25]. However, this finding should be interpreted carefully because another study reported that capsaicin cough sensitivity was higher in patients aged ≥ 50 years than in those aged < 50 years [26].

Sleep disturbance was not associated with age in any of the five etiologies, which might suggest a distinct pathophysiology of night cough. Although all the COAT items were closely interrelated, which enabled severity prediction of other aspects of cough using each item, patterns of inter-relationships between demographic and cough characteristics differed according to the etiology. Specifically, hypersensitivity to irritants in asthma and sleep disturbance in GERD were not associated with other cough characteristics. The existence of isolated symptoms in certain diseases may be a key finding to help diagnose and understand the pathophysiology of cough, although the mechanism of this finding remains to be elucidated.

This study has some limitations that should be addressed. Patients with multiple causes of chronic cough were excluded from this analysis because our study aimed to characterize cough according to etiology and compare the characteristics between different diagnoses. However, many patients have multiple causes of chronic cough, and the characteristics of cough may appear mixed in these patients, further complicating the diagnostic process. Nevertheless, our approach can be useful for understanding the pathophysiology of cough with different etiologies. Second, only non-smokers were included in the study. There were three patients in our cohort in whom smoking was determined as the cause of chronic cough, but they were excluded from the current analysis due to the small sample size. In addition, the number of patients in each group was disproportionate, which may have influenced the study results. Further studies with larger sample sizes including smokers are required. Third, given that the epidemiology of chronic cough differs among countries, different patterns could be found in other circumstances, which may limit the generalizability of the study results to other races. Further studies are required to confirm these results.

## 5. Conclusions

In conclusion, the cough severity and characteristics differed according to the cough etiology. Demographic features and cough characteristics were intercorrelated, and different patterns in their relationships were observed according to the cough etiology. Detailed evaluation of cough may help differentiate cough etiology. Further studies are necessary to elucidate the mechanisms of cough.

## Figures and Tables

**Figure 1 jcm-12-05383-f001:**
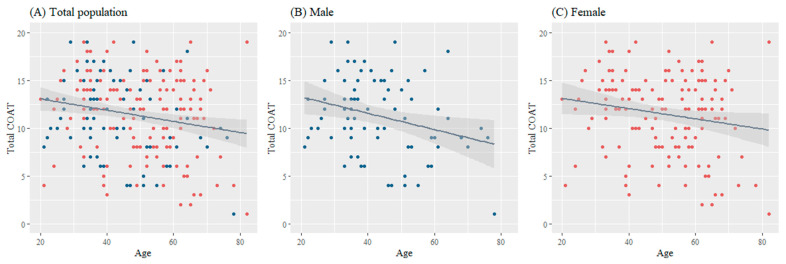
Effects of age on cough severity stratified by sex. Regression coefficient for the total population: −0.065, *p* < 0.001; men: −0.103, *p* = 0.002; women: −0.053, *p* = 0.038. Abbreviation: COAT, COugh Assessment Test.

**Figure 2 jcm-12-05383-f002:**
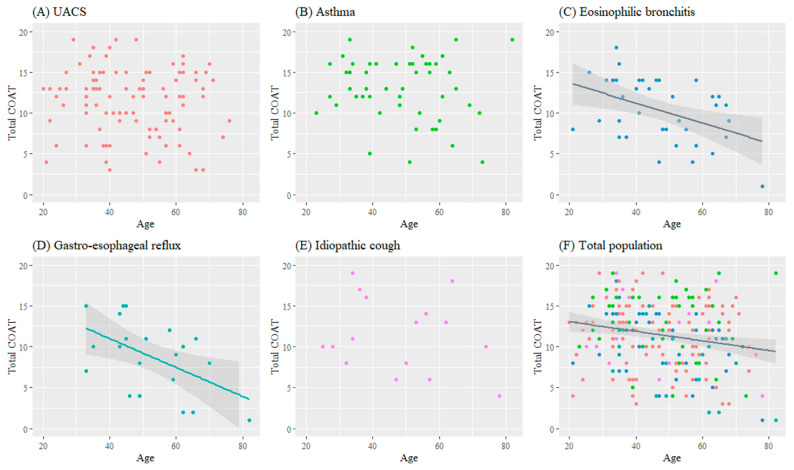
Association between age and cough severity using COAT scores according to cough etiology. Regression lines are drawn only when a significant association exists. Regression coefficient for UACS: −0.022, *p* = 0.450; asthma, −0.026, *p* = 0.520; eosinophilic bronchitis, −0.123, *p* = 0.007; GERD, −0.177, *p* = 0.016; idiopathic cough, −0.062, *p* = 0.401; total population, −0.065, *p* < 0.001. Abbreviations: COAT, COugh Assessment Test; UACS, upper airway cough syndrome.

**Figure 3 jcm-12-05383-f003:**
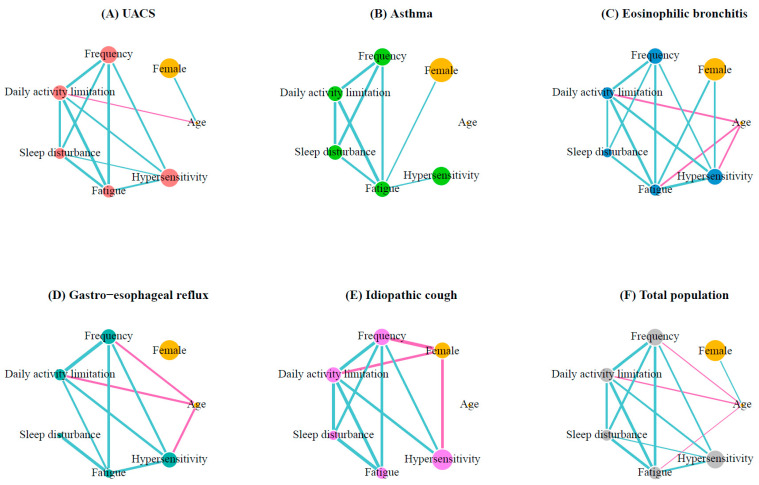
Correlation networks of demographic and cough characteristics for each cough etiology. Each variable is represented as a specific node. The size of the node indicates the prevalence: patients with an age > 65 years, women, and the relative COAT scores that are calculated by the mean of each COAT item score divided by the total COAT score. Links or edges between nodes indicate statistically significant associations (*p* < 0.05). The blue and pink lines across nodes indicate the positive and negative correlations, respectively. The thickness of an edge represents the strength of the correlation (Pearson’s R coefficient). Abbreviations: COAT, COugh Assessment Test; UACS, upper airway cough syndrome.

**Figure 4 jcm-12-05383-f004:**
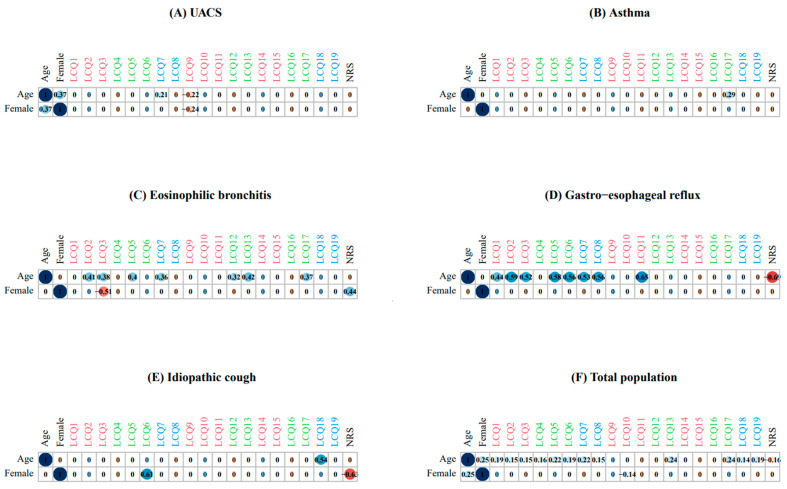
Correlation between age, sex, and K-LCQ item scores according to etiology. Physical, psychological, and social domains are represented by the colors red, green, and blue, respectively. Positive and negative correlations are indicated by the blue and red circles, respectively. The intensity of the colors reflects the magnitude of the correlation. Chest/stomach pain (LCQ1), bothersome phlegm (LCQ2), tiredness (LCQ3), feeling in control of cough (LCQ4), embarrassment (LCQ5), anxiety (LCQ6), interference with daily work (LCQ7), interference with overall life enjoyment (LCQ8), hypersensitivity to irritants (LCQ9), sleep disturbances (LCQ10), frequency of coughing bouts (LCQ11), frustration (LCQ12), feeling of being fed-up (LCQ13), about voice hoarseness (LCQ14), and loss of energy (LCQ15), worries about serious illness (LCQ16), and concerns what about other people may think (LCQ17), interruption of conversation/phone call LCQ18), and annoyance to partner/family/friends (LCQ19). Abbreviation: LCQ, Leicester Cough Questionnaire.

**Table 1 jcm-12-05383-t001:** Baseline characteristics and the COAT scores in the study patients.

	Total (*n* = 220)	UACS (*n* = 95)	Asthma (*n* = 51)	Eosinophilic Bronchitis (*n* = 37)	GERD (*n* = 21)	Idiopathic (*n* = 16)
Age, years	47.9 ± 14.0	46.7 ± 14.1	48.6 ± 13.6	47.3 ± 13.8	52.4 ± 12.9	48.0 ± 16.4
Female (%)	144 (65.5)	58 (61.1)	39 (76.5)	26 (70.3)	13 (61.9)	8 (50.0)
COAT questionnaire						
Frequency	2.62 ± 0.81	2.66 ± 0.82	2.78 ± 0.73	2.49 ± 0.73	2.33 ± 0.97	2.56 ± 0.89
Daily activity limitation	2.22 ± 1.06	2.29 ± 0.93	2.41 ± 1.12	1.95 ± 1.08	1.81 ± 1.21	2.38 ± 1.15
Sleep disturbance	1.76 ± 1.26	1.77 ± 1.23 *	2.37 ± 1.18 *	1.51 ± 1.07	0.90 ± 1.18	1.50 ± 1.32
Fatigue	2.00 ± 1.20	1.96 ± 1.18	2.51 ± 1.05 *	1.84 ± 1.21	1.38 ± 1.20	1.82 ± 1.33
Hypersensitivity	2.80 ± 1.01	2.83 ± 1.01	2.96 ± 1.00 *	2.54 ± 1.04	2.38 ± 1.20	3.19 ± 0.75 *
Total score	11.40 ± 4.16	11.52 ± 4.00 *	13.04 ± 3.79 *	10.32 ± 3.88	8.81 ± 4.38	11.44 ± 4.53
K-LCQ score	11.06 ± 3.19	10.78 ± 3.02 *	10.21 ± 3.26 *	12.10 ± 2.79	12.57 ± 3.81	11.05 ± 3.01
Physical domain	4.04 ± 0.95	3.93 ± 0.92 *	3.73 ± 0.94 *	4.42 ± 0.80	4.59 ± 1.11	4.12 ± 0.73
Psychological domain	3.45 ± 1.17	3.36 ± 1.11 *	3.22 ± 1.18 *	3.74 ± 1.10	4.00 ± 1.40	3.37 ± 1.16
Social domain	3.56 ± 1.35	3.49 ± 1.25	3.26 ± 1.41	3.94 ± 1.22	3.98 ± 1.67	3.56 ± 1.41
Cough NRS	6.04 ± 2.25	6.07 ± 2.26	6.53 ± 2.26 *	5.68 ± 1.99	5.19 ± 2.56	6.19 ± 2.14

* Indicates clinical significance (*p* < 0.05) compared to the GERD group as a reference. Abbreviations: COAT, COugh Assessment Test; UACS, upper airway cough syndrome; GERD, gastroesophageal reflux disease; K-LCQ, Korean version of the Leicester Cough Questionnaire; NRS, numeric rating scale.

**Table 2 jcm-12-05383-t002:** A correlation matrix of the demographic features and the COAT scores.

	Age	Female	COAT 1	COAT 2	COAT 3	COAT 4	COAT 5	Total COAT Score
Age	1.000	0.237 *	−0.237 *	−0.296 *	−0.025	−0.179 *	−0.154 *	−0.218 *
Female	-	1.000	−0.055	−0.085	0.084	0.125 *	0.009	0.031
COAT questionnaire								
Frequency	-	-	1.000	0.667 *	0.517 *	0.576 *	0.404 *	0.783 *
Daily activity limitation	-	-	-	1.000	0.515 *	0.666 *	0.451 *	0.839 *
Sleep disturbance	-	-	-	-	1.000	0.617 *	0.246 *	0.768 *
Fatigue	-	-	-	-	-	1.000	0.458 *	0.865 *
Hypersensitivity	-	-	-	-	-	-	1.000	0.642 *
Total COAT score	-	-	-	-	-	-	-	1.000

* Indicates statistical significance (*p* < 0.05). Abbreviations: COAT, COugh Assessment Test; COAT 1, cough frequency; COAT 2, daily activity limitation; COAT 3, sleep disturbance; COAT 4, fatigue; COAT 5, hypersensitivity to irritants.

**Table 3 jcm-12-05383-t003:** Comparison of the COAT scores between men and women.

	Male	Female	*p*-Value	Male	Female	*p*-Value
	Total			Upper airway cough syndrome		
Frequency	2.68 ± 0.79	2.59 ± 0.83	0.42	2.70 ± 0.70	2.64 ± 0.89	0.71
Daily activity limitation	2.34 ± 1.02	2.15 ± 1.08	0.20	2.51 ± 0.87	2.16 ± 0.95	0.07
Sleep disturbance	1.61 ± 1.20	1.83 ± 1.29	0.21	1.68 ± 1.16	1.83 ± 1.29	0.56
Fatigue	1.80 ± 1.22	2.11 ± 1.18	0.06	1.95 ± 1.15	1.97 ± 1.20	0.94
Hypersensitivity	2.77 ± 1.03	2.79 ± 1.02	0.89	2.84 ± 0.99	2.83 ± 1.03	0.96
Total COAT score	11.20 ± 4.16	11.47 ± 4.22	0.65	11.68 ± 3.79	11.41 ± 4.16	0.76
	Asthma			Eosinophilic bronchitis		
Frequency	2.75 ± 0.75	2.79 ± 0.73	0.85	2.18 ± 0.87	2.62 ± 0.64	0.10
Daily activity limitation	2.17 ± 1.03	2.49 ± 1.14	0.39	1.55 ± 1.04	2.12 ± 1.07	0.14
Sleep disturbance	1.92 ± 1.16	2.51 ± 1.17	0.13	1.14 ± 1.03	1.73 ± 1.04	0.06
Fatigue	1.92 ± 1.08	2.69 ± 0.98	0.02	1.00 ± 1.00	2.19 ± 1.13	<0.01
Hypersensitivity	3.08 ± 0.67	2.92 ± 1.09	0.63	2.00 ± 1.00	2.77 ± 0.99	0.04
Total COAT score	11.83 ± 3.69	13.41 ± 3.80	0.21	7.73 ± 3.44	11.42 ± 3.57	<0.01
	Gastroesophageal reflux disease			Idiopathic cough		
Frequency	2.63 ± 0.92	2.15 ± 0.99	0.29	3.25 ± 0.46	1.88 ± 0.64	<0.01
Daily activity limitation	2.38 ± 1.06	1.46 ± 1.20	0.09	3.00 ± 0.93	1.75 ± 1.04	0.02
Sleep disturbance	1.50 ± 1.31	0.54 ± 0.97	0.07	2.13 ± 1.46	0.88 ± 0.83	0.05
Fatigue	1.50 ± 1.41	1.31 ± 1.11	0.73	2.25 ± 1.49	1.38 ± 1.06	0.20
Hypersensitivity	2.50 ± 0.93	2.31 ± 1.03	0.67	3.63 ± 0.52	2.75 ± 0.71	0.01
Total COAT score	10.50 ± 4.31	7.77 ± 4.25	0.17	14.25 ± 3.69	8.63 ± 3.50	<0.01

Abbreviation: COAT, COugh Assessment Test.

**Table 4 jcm-12-05383-t004:** Multivariate analysis for the association between each cause and COAT score.

	Odds Ratio * (95% Confidence Interval)				
	UACS	Asthma	Eosinophilic Bronchitis	GERD	Idiopathic
Frequency	1.07 (0.76–1.50)	1.44 (0.95–2.18)	0.76 (0.49–1.18)	0.69 (0.40–1.20)	0.91 (0.48–1.73)
Daily activity limitation	1.08 (0.83–1.41)	1.33 (0.96–1.83)	0.72 (0.51–1.02)	0.73 (0.48–1.13)	1.19 (0.70–2.00)
Sleep disturbance	1.03 (0.83–1.27)	1.70 (1.29–2.23)	0.82 (0.61–1.10)	0.51 (0.33–0.79)	0.86 (0.56–1.31)
Fatigue	0.94 (0.75–1.18)	1.66 (1.22–2.25)	0.84 (0.62–1.14)	0.66 (0.45–0.98)	0.91 (0.59–1.40)
Hypersensitivity	1.06 (0.82–1.39)	1.27 (0.91–1.77)	0.75 (0.53–1.05)	0.72 (0.47–1.08)	1.71 (0.93–3.17)
Total COAT score	1.09 (0.94–1.08)	1.16 (1.06–1.26)	0.92 (0.84–1.00)	0.87 (0.76–0.97)	1.01 (0.89–1.15)

* Adjusted for age and sex. Abbreviations: COAT, COugh Assessment Test; UACS, upper airway cough syndrome; GERD, gastroesophageal reflux disease.

## Data Availability

The dataset generated during this study is available from the corresponding author upon reasonable request.

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
