# Peer review of "Cough Characteristics and Their Association Patterns According to Cough Etiology: A Network Analysis"

_jcm, 2023, doi:10.3390/jcm12165383_

Round 1
Reviewer 1 Report
Dear authors
The study has a cross-sectional design where patients with chronic coughs were enrolled and their demographic features and cough severity were assessed. Chronic cough can significantly impact a person's quality of life, leading to discomfort, social limitations, and psychological distress. Questionnaires help assess the impact of cough on daily life, allowing healthcare providers to address these concerns.
The study design
Your approach can provide valuable insights but has some limitations.
The study included 220 adult patients with chronic cough, which is not a large sample size for the number of variables, this is also a limitation.
This study used a correlation network analysis to examine associations between demographic features and cough severity/characteristics across different etiologies. This type of analysis can provide a comprehensive view of the relationships between variables, but the methods and assumptions of the network analysis need to be appropriately justified. There is no reference to statistical key assumptions.
Results /Discussion
You have analyzed the effect of age on different cough etiologies; however, cough due to gastroesophageal reflux disease is described to occur in more advanced age. According to your results in some way contradictory since the COAT score is inversely correlated with age., this is an important point to discuss.
In your effort to better understand the complex interplay between age, sex, cough severity, and characteristics using the COAT score for each cough etiology, correlation networks were established. However, certain relationships appear to be somewhat contradictory when compared to the results obtained from univariate or multivariate analyses, and it is imperative to address this discrepancy in the discussion.
I believe that the correlation networks approach is becoming increasingly important as it takes the information we have about the variables to another level.
Specific comments:
- Line 133- Something is wrong with the table caption, abbreviations are missing.
- Figure 4 is too small we cannot visualize the results. Please consider enlarging the images of scores.
- Please confirm the Pearson values in the matrix table, some values are not considered when <0.3.
Reviewer 2 Report
Thank you for the opportunity to read this manuscript and congratulations to the authors for their work. I highly appreciate the text submitted for review.
Here are some minor suggestions for improvement:
In lines 329-330 (page 10 of 12) it is stated that "Supplementary Materials: The following supporting information can be downloaded at: 329 www.mdpi.com/xxx/s1, Figure S1: title; Table S1: title; Video S1: title". Please, review the phrase to provide the aforementioned supplementary materials.
Author Response
Response: Thank you for your kind review and positive feedbacks. Following your comment, we have modified the "Supplementary Materials" section, as follows.
Supplementary Materials: The following supporting information can be downloaded at: www.mdpi.com/xxx/s1, Figure S1: Histograms for age distribution according to the cause of chronic cough; Figure S2: Association between age and cough severity using K-LCQ scores according to cough etiology; Figure S3: Association between age and cough severity using NRS scores according to cough etiology; Table S1: Correlation matrix showing Pearson correlation coefficients in each cough etiology.